# Regulation of Fetal Genes by Transitions among RNA-Binding Proteins during Liver Development

**DOI:** 10.3390/ijms21239319

**Published:** 2020-12-07

**Authors:** Toru Suzuki, Shungo Adachi, Chisato Kikuguchi, Shinsuke Shibata, Saori Nishijima, Yurie Kawamoto, Yusuke Iizuka, Haruhiko Koseki, Hideyuki Okano, Tohru Natsume, Tadashi Yamamoto

**Affiliations:** 1Center for Integrative Medical Sciences, Laboratory for Immunogenetics, RIKEN, Kanagawa 230-0045, Japan; chisato.kikuguchi@riken.jp (C.K.); tadashi.yamamoto@oist.jp (T.Y.); 2Molecular Profiling Research Center for Drug Discovery, National Institute of Advanced Industrial Science and Technology, Tokyo 135-0064, Japan; s.adachi@aist.go.jp (S.A.); t-natsume@aist.go.jp (T.N.); 3Department of Physiology, Keio University School of Medicine, Tokyo 160-8582, Japan; shibata@keio.jp (S.S.); hidokano@a2.keio.jp (H.O.); 4Cell Signal Unit, Okinawa Institute of Science and Technology Graduate University, Okinawa 904-0495, Japan; nishijma@oist.jp; 5Center for Integrative Medical Sciences, Laboratory for Developmental Genetics, RIKEN, Kanagawa 230-0045, Japan; yurie.kawamoto@riken.jp (Y.K.); yusuke.iizuka@riken.jp (Y.I.); haruhiko.koseki@riken.jp (H.K.)

**Keywords:** RNA binding proteins, fetal liver genes, ccr4-not complex

## Abstract

Transcripts of *alpha-fetoprotein* (*Afp*), *H19*, and *insulin-like growth factor 2* (*Igf2*) genes are highly expressed in mouse fetal liver, but decrease drastically during maturation. While transcriptional regulation of these genes has been well studied, the post-transcriptional regulation of their developmental decrease is poorly understood. Here, we show that shortening of poly(A) tails and subsequent RNA decay are largely responsible for the postnatal decrease of *Afp*, *H19*, and *Igf2* transcripts in mouse liver. IGF2 mRNA binding protein 1 (IMP1), which regulates stability and translation efficiency of target mRNAs, binds to these fetal liver transcripts. When IMP1 is exogenously expressed in mouse adult liver, fetal liver transcripts show higher expression and possess longer poly(A) tails, suggesting that IMP1 stabilizes them. IMP1 declines concomitantly with fetal liver transcripts as liver matures. Instead, RNA-binding proteins (RBPs) that promote RNA decay, such as cold shock domain containing protein E1 (CSDE1), K-homology domain splicing regulatory protein (KSRP), and CUG-BP1 and ETR3-like factors 1 (CELF1), bind to 3′ regions of fetal liver transcripts. These data suggest that transitions among RBPs associated with fetal liver transcripts shift regulation from stabilization to decay, leading to a postnatal decrease in those fetal transcripts.

## 1. Introduction

Genes that are expressed in fetal and neonatal stages, but inactivated in adults, participate in early developmental processes and maintenance of immature tissue characteristics. Some of these genes are reactivated during tumorigenesis, tissue injury, or regeneration; hence, they are called oncofetal genes [1]. *Afp* and *H19* genes show an oncofetal expression pattern in mammalian liver. The *Igf2* gene also shows an oncofetal expression pattern in mice, but not in humans [2].

AFP is an abundant serum protein in mammalian fetuses; however, reactivation of *Afp* expression is observed in ~80% of hepatocellular carcinoma (HCC) patients [3]. *H19* is a long non-coding RNA responsible for several liver functions and pathogenesis [4], whereas *Igf2* encodes a growth factor for a transmembrane receptor, transducing signals for cell growth, survival, and migration during tissue development [5]. Epigenetic abnormalities in both genes are detected in HCC [6]. Moreover, an increase of these genes is associated with other types of liver diseases, such as hepatitis and alcohol-dependent liver injury [7].

The postnatal decrease of fetal liver transcripts has been analyzed mainly at the transcriptional level. FOXA1 is a critical transcriptional activator of *Afp* and *H19* in liver [8,9]. Direct binding of p53 to the *Afp* promoter induces histone modification for transcriptional repression, resulting in suppression of FOXA1-mediated *Afp* transcription [8,10]. In addition, a developmental decrease of FOXA1 binding to the H19 enhancer is observed [9]. Insufficient postnatal reduction of AFP observed in BALB/cJ mice is due to a natural hypomorph mutation in the *Zhx2* gene locus [11,12]. ZHX2 functions as a transcriptional repressor [13], and exogenous expression of the *Zhx2* gene significantly reduces *Afp* and *H19* transcripts [12]. ZHX2 level increases after birth, partly accounting for postnatal decrease of *Afp* and *H19* transcripts. Liver-specific suppression of the zinc finger protein, ZBTB20, leads to persistent *Afp* expression in liver throughout adult life, indicating that ZBTB20 also contributes to postnatal *Afp* repression [14]. However, it seems unlikely that these transcriptional mechanisms fully account for postnatal repression of fetal liver transcripts. Indeed, involvement of post-transcriptional mechanisms is suggested in ZHX2-mediated reduction of *Afp* and *H19* expression [15].

Among post-transcriptional mechanisms, translational repression and RNA decay are relevant to decreased mRNA levels. MicroRNAs (miRNAs) are major translational repressors, which form RNA-induced silencing complexes (RISC) and bind mainly to 3′ untranslated regions (3′UTRs) [16]. Following translational repression, mRNAs usually undergo degradation in a manner dependent upon ribonucleases, including the CCR4-NOT complex. The CCR4-NOT complex initiates decay of mRNAs by shortening their poly(A) tails. To recognize target transcripts, the CCR4-NOT complex uses various RBPs, such as CSDE1, bicaudal-C, RISC, tristetraproline, butyrate response factors 1 and 2 (BRF1/2), Nanos2, Roquin, and dead end 1 [17,18,19,20,21,22,23,24].

RBPs compete with one another for binding and regulation of common target RNAs during various biological events. While Hu antigen D (HuD) stabilizes and promotes translation of *GAP-43* mRNA, KSRP binding to *GAP-43* mRNA leads to destabilization of the mRNA [25,26]. Both HuD and KSRP recognize AU-rich element (ARE) in *GAP-43* mRNA, and KSRP can displace HuD from *GAP-43* mRNA [25]. CELF1 competes with HuR or calreticulin for translation of *occluding* or *p21* mRNAs, respectively [27,28]. Importantly, the effect of competitive binding was clearly reflected in axonal outgrowth, proliferation, or other cellular functions [26,27,28].

When mice lack expression of *microRNA-122* (an abundant liver miRNA), Dicer, which produces mature miRNAs, or core subunits of the CCR4-NOT complex in liver, enhanced expression of *Afp*, *H19*, and *Igf2* transcripts is detected in adult livers compared to corresponding controls [29,30,31,32,33], suggesting that post-transcriptional mechanisms are responsible for the postnatal decrease of fetal liver transcripts. On the other hand, deletion of *miR-122*, Dicer, or the CCR4-NOT complex results in severe liver damage [29,30,31,32,33]. More detailed analyses are required to determine whether abundant expression of *Afp*, *H19*, and *Igf2* in adult livers of these mouse strains is due to loss of post-transcriptional control mechanisms or to secondary effects of liver damage.

In this study, we provide evidence that CCR4-NOT complex-mediated RNA decay is involved in the postnatal decrease of *Afp*, *H19*, and *Igf2* transcripts. Our analyses also show that different RBPs bind to 3′ regions of *Afp*, *H19*, and *Igf2* transcripts, depending on liver developmental stages. Based on our findings, we propose a model for developmental repression of oncofetal genes involving interactions between RBPs and RNAs.

## 2. Results

### 2.1. Afp, H19, and Igf2 Transcripts Decrease in a CNOT3-Dependent Manner as Liver Matures

We have previously shown that *Afp*, *H19*, and *Igf2* transcripts are more highly expressed and had longer poly(A) tails in livers from adult mice lacking *Cnot3* specifically in liver (*Cnot3-LKO* mice) than in those from control mice possessing *floxed* alleles in the *Cnot3* gene locus (called control mice hereafter) [32]. In this report, we use *Afp*, *H19*, and *Igf2* transcripts as representative fetal liver transcripts in mice. We examined expression and poly(A) tail lengths of fetal liver transcripts in mouse liver at various postnatal days. At 3 days and 1 week after birth, *Afp* transcripts were expressed at similar levels in livers from control and *Cnot3-LKO* mice. *Igf2* and *H19* transcripts showed higher expression in livers from control mice than those from *Cnot3-LKO* mice (Figure 1A). In control mouse livers, levels of these three transcripts started to decrease 2 weeks after birth, and declined steadily thereafter (Figure 1A). Concomitantly, poly(A) tails of *Afp* and *Igf2* transcripts became shortened (Figure 1B). Poly(A) tails of *H19* transcripts were elongated 1 week after birth compared to embryonic day 16, and were gradually shortened, as were those of *Afp* and *Igf2* transcripts (Figure 1B). The decrease of fetal liver transcripts in livers from *Cnot3-LKO* mice was significantly less than that observed in livers from control mice (Figure 1A). Consistent with this, fetal liver RNAs had longer poly(A) tails in livers from *Cnot3-LKO* mice than those from control mice (Figure 1B). These results suggest that CCR4-NOT complex-mediated poly(A) shortening and subsequent RNA decay is involved in postnatal decrease of fetal liver transcripts.

### 2.2. IMP1 Decreases Concomitantly with Fetal Liver Transcripts during Liver Development

As mice grew from 1 week to 5 weeks, subunits of the CCR4-NOT complex were expressed at more or less constant levels in the liver (Figure 2A). Furthermore, immunoprecipitation analyses showed that a comparable amount of the CCR4-NOT complex was continuously present during this developmental period (Figure 2B). Therefore, the postnatal decrease of fetal liver transcripts was not determined by the abundance of the CCR4-NOT complex.

Then, we examined expression of molecules that bind to and stabilize RNAs. IMP family proteins regulate RNA at various post-transcriptional levels, determining localization, translation, or stability [34,35]. Among them, IMP1 and IMP3 are highly expressed in mouse embryonic stages, but show very low expression in most adult tissues [34,36]. We detected a decrease of IMP1 protein in liver as mice grew from 1 week to 5 weeks after birth both in the presence or the absence of CNOT3 (Figure 2C). Phosphorylation state of IMP1 was not influenced by CNOT3 suppression, either (Figure 2C). IMP1 and AFP started to decrease at about 2 weeks after birth, and both IMP1 and AFP were barely detectable 3 weeks after birth and thereafter (Figure 2C). The decrease of fetal liver transcripts and shortening of their poly(A) tails were particularly noticeable at around 2 weeks after birth (Figure 1). These data suggest that IMP1 maintains expression and translation of fetal liver transcripts during fetal and perinatal days, and that a decrease of IMP1 enables the CCR4-NOT complex to degrade mRNAs at a later developmental stage. RNA immunoprecipitation experiments demonstrated binding of IMP1 to fetal liver transcripts, further supporting the idea (Figure 2D). Like IMP1, IMP3 decreased during liver maturation, though the magnitude of the decrease was less prominent than that of IMP1 (Figure 2C). However, IMP3 did not decrease in livers from *Cnot3-LKO* mice in contrast to IMP1 (Figure 2C). We confirmed that both IMP3 mRNA and protein were more highly expressed in livers from *Cnot3-LKO* mice than those from control mice at 4 weeks of age (Figure 3A,B), raising the possibility that high expression of fetal liver transcripts is due to persistent expression of IMP3.

### 2.3. High Expression of Fetal Liver Transcripts in Adult Cnot3-LKO Mice Is not Affected by the Absence of IMP3

We first examined whether increased IMP3 is responsible for continuous expression of fetal liver transcripts in adult livers from *Cnot3-LKO* mice. We generated mice lacking the *Imp3* gene (*Imp3^−/−^* mice) and crossed them with *Cnot3-LKO* mice to obtain mouse strains lacking both *Imp3* and *Cnot3* genes in liver (*Imp3^−/−^*; *Cnot3-LKO* mice). Even in the absence of IMP3 in all tissues, mice were born at Mendelian frequency (*Imp3^+/+^*:*Imp3^+/−^*:*Imp3^−/−^* = 9:14:7 in 4 crossings) and grew without any obvious abnormalities (Appendix A). *Imp3^−/−^* mice indeed showed normal liver histology (Appendix A). When we examined AFP protein expression in livers from 4-week-old mice, AFP was detected in livers from *Cnot3-LKO* and *Imp3^−/−^*; *Cnot3-LKO* mice, but not those from control and *Imp3^−/−^* mice (Figure 3C). Moreover, fetal liver transcripts were more highly expressed and had longer poly(A) tails in livers from *Imp3^−/−^*; *Cnot3-LKO* mice compared to those from *Imp3^−/−^* mice (Figure 3D,E). These results suggested that an increase of IMP3 in livers from *Cnot3-LKO* mice is dispensable or irrelevant to postnatal expression of fetal liver transcripts. It should be also noted that IMP3 increase has little effect on phenotypes of *Cnot3-LKO* mice, because we observed comparable abnormalities in *Imp3^−/−^*; *Cnot3-LKO* mice to those we have previously observed in *Cnot3-LKO* mice, such as decreased liver and body weights, misaligned sinusoids, and inflammation (Appendix A, and [32]).

### 2.4. Poly(A) Tail Elongation and Increased Levels of Fetal Liver Transcripts in Imp1-Transgenic Mice

Next, to investigate the relationship between the developmental decrease of IMP1 and expression of fetal liver transcripts, we generated mice possessing an *Imp1* transgene under control of the *Albumin* enhancer and promoter (*Imp1*-transgenic mice), so that IMP1 was expressed in liver even after it matured. We confirmed that IMP1 was expressed in livers from *Imp1*-transgenic mice, but not in livers from wild-type mice at 4 weeks of age (Figure 4A). *Imp1*-transgenic mice were almost normal and showed comparable body and liver weights to those of wild-type mice (Appendix A). Liver histology was also normal at the ages we examined (Appendix A). In spite of normal appearance, *Afp*, *Igf2* and *H19* transcripts showed significantly higher expression in livers from *Imp1*-transgenic mice than those from wild-type mice at 4 weeks of age (Figure 4B). Furthermore, *Afp* and *Igf2* transcripts had longer poly(A) tails in liver from *Imp1*-transgenic mice (Figure 4C). Whereas expression of unspliced immature transcripts of *Afp* and *Igf2* genes was comparable in livers between *Imp1*-transgenic mice and wild-type mice, the mature/immature ratios of transcripts increased significantly in *Imp1*-transgenic mice, suggesting that those transcripts were stabilized (Figure 4D,E). Poly(A) tail elongation of *H19* transcripts was less prominent than of *Afp* and *Igf2* transcripts (Figure 4C). *H19* transcripts undergo splicing like that of protein-coding RNAs [37], leading us to examine the level of unspliced immature *H19* transcripts. Immature *H19* transcripts increased significantly in *Imp1*-transgenic mice (Figure 4D); therefore, IMP1 expressed from the transgene appeared to influence mainly transcription of the *H19* gene. These data suggest that IMP1, at least in part, is responsible for developmental expression of fetal liver genes.

### 2.5. Binding of RBPs to Fetal Liver Transcripts Changes during Liver Maturation

Increased expression and poly(A) tail elongation of fetal liver transcripts upon exogenous IMP1 expression led us to speculate that RBPs responsible for RNA decay, contribute to the postnatal decrease of fetal liver transcripts. We performed an in vitro RNA binding experiment that we previously established [20]. We prepared FLAG-peptides conjugated with approximately 500 bases of 3′ regions of fetal liver transcripts and examined binding proteins following incubation of FLAG-RNAs with liver lysates from 2 or 3-week-old mice.

We first focused on RBPs that recognize AREs, because the nucleotide sequences are usually in the 3′ UTRs of their mRNAs and function critically as cis-acting elements in mRNA decay [38,39]. Immunoblot analysis showed that KSRP bound to 3′ regions of fetal liver transcripts (Figure 5A, top). Binding was enhanced in lysates from 3-week-old mice compared to those from 2-week-old mice, especially in *Igf2* and *H19* transcripts. In contrast, BRF1/2 and AU-rich element RNA binding factor 1 (AUF1) failed to bind to those transcripts (Figure 5A). CELF1, which recognize GU-rich elements (GREs) or U-rich elements in addition to AREs [40,41], bound to 3′ regions in *H19* and *Igf2* transcripts, and binding similarly increased in livers from 3-week-old mice (Figure 5A). CSDE1 interacts with the CCR4-NOT complex and promotes mRNA decay [17]. CSDE1 bound predominantly to 3′ regions of *Afp* and *Igf2* transcripts in livers from 3-week-old mice (Figure 5A). IMP1 bound to all the three transcripts and the amount of FLAG-RNA-bound IMP1 was reduced as the level of IMP1 protein decreased in livers from 2-week-old to 3-week-old mice (Figure 5A).

We confirmed that binding of RNA decay-related RBPs to fetal liver transcripts was enhanced as mice matured using competition experiments. We first incubated FLAG-RNAs with liver lysates from adult mice (4~5 weeks of age) and collected complexes of FLAG-RNAs and associated proteins using anti-FLAG antibody. Subsequently, anti-FLAG immunoprecipitates were incubated with liver lysates from 1-week-old mice. Proteins binding to FLAG-RNAs after the second incubation were analyzed by immunoblot (Figure 5B). CSDE1 bound to the 3′ regions of *Afp* and *Igf2* transcripts in liver lysates from 4-week-old mice, and binding was reduced following incubation with liver lysates from 1-week-old mice (Figure 5C). Similar results were obtained when we examined binding of KSRP or CELF1 to 3′regions of *H19* or *Igf2* transcripts (Figure 5C). Instead of reduced binding of those proteins to FLAG-RNAs, binding of IMP1 was evident after incubation with liver lysates from 1-week-old mice (Figure 5C).

## 3. Discussion

Liver maturation requires an increase of liver function-related genes, as well as a decrease of genes related to immature liver. A strictly controlled network of transcription factors largely governs liver maturation and function through transcriptional activation or repression [42,43]. This study reveals contributions of post-transcriptional mechanisms to postnatal repression of fetal liver genes. While one of the major RNA decay molecules, the CCR4-NOT complex, is continually present during liver development, different RBPs bind to 3′ regions of fetal liver transcripts, depending on the developmental stage. RNA decay-related RBPs, such as KSRP, CELF1, and CSDE1 bound more abundantly to fetal liver transcripts at later developmental stages (Figure 5A). This increased binding paralleled the developmental decrease of IMP1 protein, identifying IMP1 as one of the candidate RBPs that competitively regulate fetal liver transcript levels. This was supported by results using transgenic mice and competitive experiments (Figure 4 and Figure 5C). Thus, we provide a model in which regulation of fetal liver transcripts shifts from stabilization to decay through a liver maturation-dependent switch of associated RBPs (Figure 6).

Exogenous IMP1 expression led to increased expression and poly(A) tail elongation in fetal liver transcripts (Figure 4). The IMP1-dependent increase of those transcripts was much less than the difference in their expression levels between neonatal and adult livers (Figure 1 and Figure 4). Similarly, poly(A) tail elongation of those transcripts in *Imp1*-transgenic was less pronounced than that in *Cnot3-LKO* mice (Figure 4, [32]). Given that IMP1 is mainly involved in post-transcriptional mechanisms [34,35], transcriptional repression of fetal liver genes should function normally in *Imp1*-transgenic mice. It is also possible that IMP1 requires cooperation of other RBPs to efficiently stabilize fetal liver transcripts, as IMP1 and HuD cooperate in binding to target mRNAs in neuronal cells [44,45]. Liver development was apparently normal even despite sustained IMP1 expression and a subsequent increase of fetal liver transcripts (Figure 4). That would be also explained by partial effects of IMP1 expression.

CELF1 is involved in many post-transcriptional processes including splicing, stabilization/translation, or degradation of transcripts [41,46]. IMP1 mediates destabilization of long non-coding RNA in a CCR4-NOT complex-dependent manner [47]. While HuR stabilizes *MyoD*, *Myogenin*, and *p21* mRNAs in muscle cells, HuR destabilizes *nucleophosmin* mRNA in cooperation with KSRP [48]. These data indicate that RBPs can have opposite effects, stabilization or destabilization of target RNAs, depending on the context, including associating partners and biological events. Further analyses are necessary to define the biological significance of competitive binding of IMP1 with KSRP, CELF1, and CSDE1 during liver development. In addition, many other RBPs except for those examined in this study, probably contribute to developmental regulation of fetal liver transcripts. Mass spectrometry analysis of FLAG-RNA bound proteins is useful to identify critical RBP groups critical for regulation [20].

RBPs antagonize miRNA-mediated post-transcriptional regulation. For example, HuR competes with various miRNAs, including miR-122, miR-548c-3p, miR-494, miR-331-3p and miR-16 for binding to 3′UTRs of target mRNAs [49,50,51,52,53,54]. IMP1 prevents the AGO2-miR-183 complex from binding to the mRNA that encodes beta-transducin repeat-containing protein 1, though the competition occurs in the coding region [55]. We have previously detected AGO2 as a FLAG-RNA-bound protein, suggesting that miRNAs associate with FLAG-RNAs [56]. Whether competition between RBPs and miRNAs is involved in regulation of fetal liver transcripts will be examined using the FLAG-RNA system in a future study.

Transcriptional activation largely contributed to an increase of *H19* transcripts in livers from *Imp1*-transgenic mice and *Cnot3-LKO* mice (Figure 4, [32]). These results are likely to be relevant to roles of the CCR4-NOT complex in transcription, in addition to its function as a deadenylase. The CCR4-NOT complex is involved in direct and indirect transcriptional regulation. In *S. cerevisiae*, the CCR4-NOT complex interacts with TATA-binding protein (TBP), TBP-associated factor, and RNA polymerase II complexes to control transcription initiation and elongation [43,44,45]. The CCR4-NOT complex regulates transcription in mammals through association with various transcriptional cofactors such as histone deacetylase, retinoid X receptor alpha, estrogen receptor, and NRC-interacting factor [24,32,46,47,48]. Our previous studies provided evidence that suppression of the CCR4-NOT complex in liver and pancreatic-cells promotes stabilization of mRNAs encoding transcription factors, leading to transcriptional activation of their target genes [23,50].

The effect of IMP1 on poly(A) tail elongation was less pronounced in *H19* transcripts than in *Afp* and *Igf2* mRNAs (Figure 4C). Given that H19 functions as an RNA in biological events, such as proliferation, drug-resistance, or tumorgenicity [4,57,58], interactions of *H19* with RBPs are not only relevant to stability, but also to functions of *H19* transcripts. *H19* regulates transcription of miR-200 through interaction with the hnRNP U/PCAF/RNA Pol II complex [59]. In particular, KSRP interacts with *H19* transcripts to destabilize *Myogenin* mRNA during myogenic differentiation [60], suggesting that the KSRP-*H19* interaction we observed in liver, is involved in regulation of other transcripts.

Upregulation of fetal liver transcripts is frequently observed in injured livers, hepatocellular carcinoma, and regenerating livers [3,61,62]. Both transcriptional and post-transcriptional mechanisms are responsible for the increase. We found that different RBPs bound to fetal liver transcripts, depending on the state of liver maturation, allowing their postnatal decrease. This facilitates studies examining whether a change in associated RBPs contributes to upregulation of fetal liver transcripts and other disease-related transcripts in various liver diseases. Manipulation of associating RBPs may provide useful information for developing novel therapeutic strategies.

## 4. Materials and Methods

### 4.1. Vectors

*Imp1* cDNA was isolated from a mouse liver cDNA pool by polymerase chain reaction (PCR) using Phusion (Thermo Fisher Scientific, Wilmington, MA, USA). The nucleotide sequence for a FLAG tag was included in the PCR primer. The amplified cDNA fragment was inserted into BamHI and XhoI sites in multicloning sites of pLIVE vector (MIR5420, Mirus Bio, Madison, WI, USA). Oligonucleotides containing loxP sequence were inserted into both BglII and XhoI sites. To construct an *Imp3*-targeted vector, a 4.0 kb genomic fragment in front of exon 1 and a 3.3 kb genomic fragment behind exon 2 were subcloned for the 5′ homology arm and 3′ homology arm, respectively. A *lacZ* fragment following a splicing acceptor and an internal ribosome entry sequence, and *neo* cassette flanked by two loxP sites were inserted between 5′ homology arm and 3′ homology arm. A thymidine kinase expression cassette was included in front of the 5′ arm for negative selection.

### 4.2. Mice

Mice carrying the floxed allele of *Cnot3* crossed with *Albumin*-*Cre* mice (#003574, The Jackson Laboratory, Bar Harbor, ME, USA) have been described previously [32]. To generate *Imp1*-transgenic mice, the BglII-PstI fragment (around 3.6 kb) containing mouse *AFP* enhancer II, mouse minimal *Albumin* promoter and FLAG-*Imp1* cDNA (see Vectors), was injected into pronuclei of fertilized eggs obtained from C57BL/6J females (CLEA Japan Inc., Tokyo, Japan). Microinjected eggs were implanted into the oviducts of plugged pseudopregnant ICR females. To generate mice lacking the *Imp3* gene, exons 1 and 2 were replaced with the *LacZ*-*neomycin resistant gene* cassette by homologous recombination (Appendix A). The linearized targeting vector was electroporated into ES cells (129/Sv). The cells were cultured in the presence of G418. We injected positive clones into C57BL/6J blastocysts and mated chimeric offspring with C57BL/6J mice (SLC Japan Inc., Shizuoka, Japan). After backcrossing heterozygous mutants with C57BL/6J mice, the F7-8 progeny of heterozygous intercrosses were used for this study. Experiments were performed according to animal use guidelines issued by the Committee of Animal Experiments at Okinawa Institute of Science and Technology Graduate University and Keio University, and Institutional Animal Care and Use Committee of RIKEN Yokohama Branch.

### 4.3. Histology

Dissected livers were fixed with 10% formaldehyde. Fixed livers were processed as paraffin embedded sections (4 μm) for hematoxylin and eosin staining. Hematoxylin 3G (8656) and eosin (8659) were from Sakura Finetek Japan (Tokyo, Japan). We captured images using a BZ X-700 microscope (Keyence, Osaka, Japan).

### 4.4. Antibodies and Reagents

Antibodies against CNOT3, CNOT6L, and CNOT8 have been described previously [56]. Anti-CNOT1 antibody (14276-1-AP) was purchased from Proteintech (Rosemont, IL, USA). Antibodies against CNOT7 (H00029883-M01) were from Abnova (Taipei, Taiwan). Antibodies against CNOT2 (#34214) and BRF1/2 (#2119) were purchased from Cell Signaling Technology (Danvers, MA, USA). Antibodies against IMP1 (RN007P) and IMP3 (RN009P) were from MBL (Nagoya, Japan). Antibodies against AFP (ab46799), AUF1 (ab61193), CELF1 (ab9549), and CSDE1 (ab201688) were obtained from Abcam (Cambridge, UK). Anti-Albumin antibody (sc-46291) was from SantaCruz (Dallas, TX, USA). Anti-KSRP antibody (A302-021A) was from Bethyl Laboratories (Montgomery, TX, USA). Antibody against FLAG-tag was from Sigma (St. Louis, MO, USA, F1804). Normal rabbit immunoglobulin G (IgG) or mouse IgG were from WAKO (Osaka, Japan, 148-09551 or 140-09511, respectively). Polyclonal antiserum against Ser181-phosphorylated IMP1 was generated by immunizing a rabbit with the synthetic peptide CQPRQGS(PO4)PVAAG coupled to KLH by their N-terminal cysteine. After depleting antibodies reactive with the corresponding non-phosphorylated peptide, phospho-specific antibodies were affinity-purified with the immunizing peptide-conjugated column. Immunizations, enzyme-linked immuno sorbent assays, and serum collection were performed at Oriental Yeast. Co., Ltd. (Tokyo, Japan).

### 4.5. Immunoprecipitation and Immunoblot Analysis

Livers were homogenized with a glass homogenizer in lysis buffer A (1% NP-40, 50 mM Tris–HCl [pH 7.5], 150 mM NaCl, 1 mM EDTA, 1 mM phenylmethylsulfonylfluoride, 10 mM NaF, 10 mM -glycerophosphate). After centrifugation at 15,000 rpm for 10 min, supernatants were filtered with a sterile filter unit (Millex-HV, 0.45 μm, Millipore, Burlington, MA, USA). Liver lysates (2 mg) were subjected to immunoprecipitation using anti-CNOT3 antibody (1 μg) and 12 microL of protein G Sepharose (GE Healthcare, Chicago, IL, USA). Immunoprecipitates or equal amounts of proteins were resolved on 8.5% SDS-polyacrylamide gels and transferred to Immobilon-p (Millipore). Membranes were blocked with 3% nonfat dry milk (Morinaga, Tokyo, Japan) in Tris-buffered saline containing 0.05% Tween 20 (TBST) for 2 h at RT. Membranes were incubated with the indicated primary antibodies overnight at 4 °C. We used Can Get Signal^®^ Immunoreaction Enhancer Solution (TOYOBO, Osaka, Japan) for antibody dilution. Membranes were washed with TBST, 3 times for 10 min at RT, and then incubated with secondary antibodies for 1 h at RT. Again, membranes were washed with TBST, 3 times for 10 min at RT. We used a Western Lightning Plus ECL (PerkinElmer, Waltham, MA, USA) for developing membranes and images were detected on an Amersham Imager 600 (GE Healthcare). We used Image J (https://imagej.net/) to measure band intensities. We performed at least two independent experiments using different samples.

### 4.6. Binding Analysis between 3′UTRs of MRNAs and Protein

cDNA was synthesized from total liver RNA. 3′ regions of *Afp*, *Igf2*, and *H19* RNA were isolated by PCR. The regions are as follows. *Afp*: 434 bases (1581-2014 in NM_007423.4), *Igf2*: 416 bases (3602-4017 in NM_010514.3), *H19*: 437 bases (2160-2596 in NR_001592.1). FLAG peptide-conjugated RNA was generated as previously described [20]. Livers were lysed with lysis buffer A. Lysates (2 mg) were incubated with FLAG-RNA (10 pmol) for 2 h. Binding proteins were purified with anti-FLAG antibody (M2)-conjugated agarose (A2220, Sigma) and analyzed by immunoblot. In competition experiments, liver lysates (2 mg) from 4 or 5-week-old mice were incubated with FLAG-RNA (10 pmol) for 2 h. Binding proteins were purified with anti-FLAG antibody-conjugated agarose. Purified FLAG-RNA-bound protein complexes were further incubated with (1) liver lysates (2 mg) from 1-week-old mice, (2) those (2 mg) from 5-week-old mice, or (3) lysis buffer only. Binding proteins were purified again with anti-FLAG antibody (M2)-conjugated agarose and analyzed by immunoblot. We performed two independent experiments using different samples.

### 4.7. RNA Analysis

Total RNA (1 μg) was used for reverse transcription with oligo(dT)12-18 primers (Thermo Fisher Scientific) using the SuperScript III First-Strand Synthesis System (Thermo Fisher Scientific). Quantitative PCR (qPCR) reactions were carried out using TB Green Premix Ex Taq II (Takara, Shiga, Japan) and the StepOnePlus Real-Time PCR System (Thermo Fisher Scientific). *Gapdh* mRNA level was used for normalization. To compare poly(A) tail lengths, we used a Poly(A) Tail-Length Assay Kit (Thermo Fisher Scientific) according to the manufacturer’s protocol. For RNA immunoprecipitation-RT-PCR (or qPCR) analysis, liver lysates were immunoprecipitated with anti-IMP, or normal rabbit IgG antibodies. RNAs in immunoprecipitates were isolated using ISOGENII (Nippon Gene, Tokyo, Japan). Primers for RT-PCR, qPCR reactions and poly(A) tail analyses are listed in Appendix A.

### 4.8. Statistical Analyses

Differences between groups were examined for statistical significance using Student’s *t*-test (two-tailed distribution with two-sample equal variance). Values represent means ± standard error of means (sem) and are represented as error bars. A *p*-value of <0.05 was considered statistically significant. No asterisk or “ns” mean not significant.

## Figures and Tables

**Figure 1 ijms-21-09319-f001:**
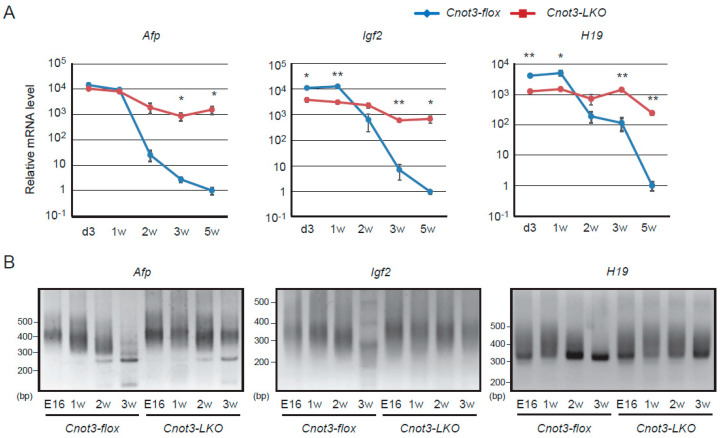
*Afp*, *H19* and *Igf2* transcripts maintain long poly(A) tails and fail to undergo postnatal decrease in the absence of CNOT3. (**A**) qPCR analysis of *Afp*, *H19*, and *Igf2* transcripts in livers from control and *Cnot3-LKO* mice at the indicated postnatal days. *Gapdh* mRNA levels were used for normalization. Levels in 5-week-old (5 w) control mice are set to 1 (*n* = 3). Values in graphs represent means ± sem. * *p* < 0.05, ** *p* < 0.01. (**B**) Comparison of poly(A) tail lengths of liver transcripts from control and *Cnot3-LKO* mice at the indicated postnatal days. Total RNA was subjected to PCR-based analysis (see Methods).

**Figure 2 ijms-21-09319-f002:**
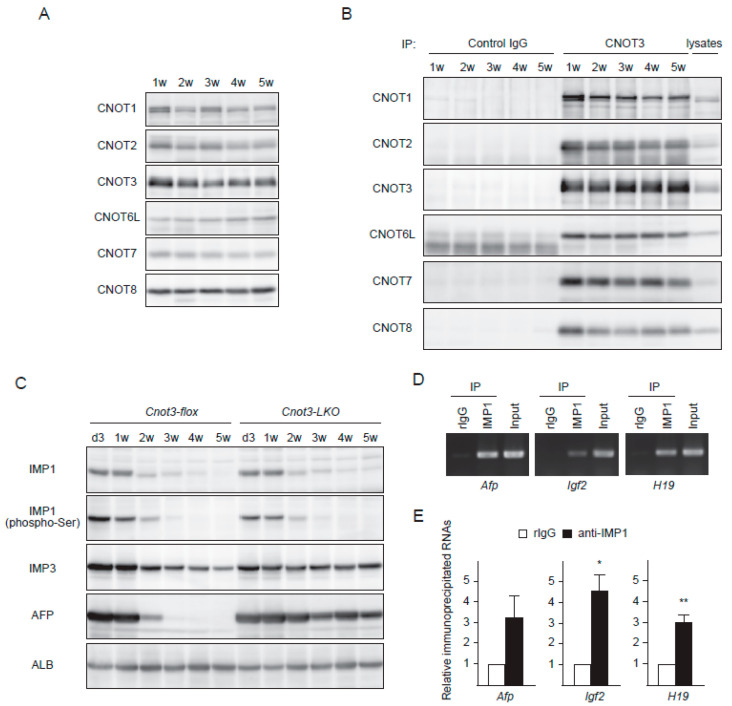
IMP1 abundance, but not that of the CCR4-NOT complex decreases as liver matures. (**A**,**B**) Immunoblot of mouse liver lysates (**A**) and anti-CNOT3 immunoprecipitates from mouse liver lysates (**B**) at the indicated postnatal days. Antibodies used are indicated on the left. (**C**) Immunoblot of liver lysates from control and *Cnot3-LKO* mice at the indicated postnatal days. Antibodies used are indicated on the left. (**D**,**E**) Liver lysates from 2 w mice were immunoprecipitated with control rabbit immunoglobulin (rIgG) or antibodies against IMP1. RNAs in immunoprecipitates were analyzed by RT-PCR (D) or qPCR (*n* = 3) (E) for the indicated genes. Values in graphs represent means ± sem. * *p* < 0.05, ** *p* < 0.01.

**Figure 3 ijms-21-09319-f003:**
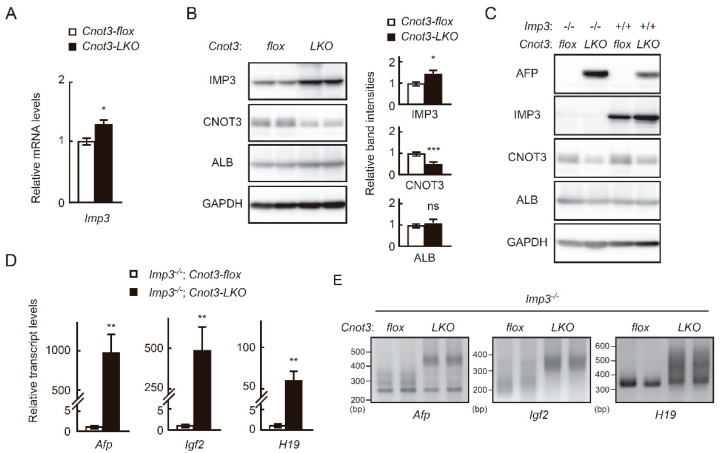
Increased levels and elongated poly(A) tails of *Afp*, *H19* and *Igf2* transcripts in *Cnot3-LKO* mice are not affected by an absence of IMP3. (**A**) qPCR of *Imp3* mRNA in livers from 4 w control and *Cnot3-LKO* mice. *Gapdh* mRNA levels were used for normalization. The *Imp3* level in control mice was set to 1 (*n* = 3). Values in graphs represent means ± sem. * *p* < 0.05. (**B**) Immunoblot of indicated molecules in livers from 4 w control and *Cnot3-LKO* mice (two representatives of three different samples). Right graphs show quantification of the immunoblot data. Relative band intensities normalized to GAPDH levels are calculated (*n* = 3). Values in control mice are set to 1. Data are presented as means ± sem. * *p* < 0.05, *** *p* < 0.001. (**C**) Immunoblot of indicated molecules in livers from control, *Imp3^−/−^*; *Cnot3-flox*, *Cnot3-LKO*, and *Imp3^−/−^*; *Cnot3-LKO* mice. (**D**) qPCR analysis of the indicated transcripts in livers from 4 w *Imp3^−/−^*; *Cnot3-flox* and *Imp3^−/−^*; *Cnot3-LKO* mice. *Gapdh* mRNA levels were used for normalization. Levels in *Imp3^−/−^*; *Cnot3-flox* mice are set to 1 (*n* = 3). Values in graphs represent means ± sem. ** *p* < 0.01. (**E**) Comparison of poly(A) tail lengths of the indicated transcripts in livers from 4 w *Imp3^−/−^*; *Cnot3-flox* and *Imp3^−/−^*; *Cnot3-LKO* mice.

**Figure 4 ijms-21-09319-f004:**
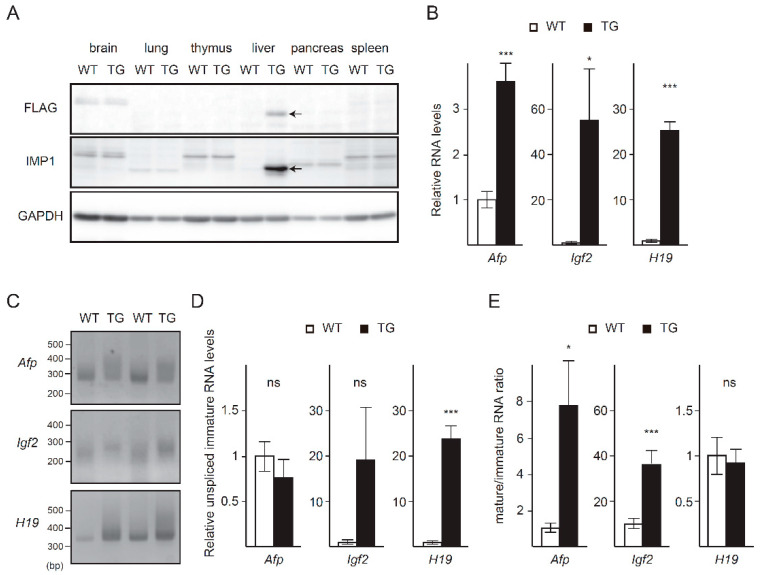
Mice exogenously expressing IMP1 in adult liver show higher expression and longer poly(A) tails of fetal liver transcripts. (**A**) Immunoblot of the indicated molecules in tissues from 4 w wild-type (WT) and *Imp1*-transgenic (TG) mice. (**B**) qPCR analysis of the indicated transcripts in livers from 4 w wild-type and *Imp1*-TG mice (*n* = 8). *Gapdh* mRNA levels were used for normalization. (**C**) Comparison of poly(A) tail lengths of the indicated transcripts in livers from 4 w WT and *Imp1*-TG mice. (**D**) qPCR analysis of unspliced immature transcripts of fetal liver genes in livers from 4 w WT and *Imp1*-TG mice (*n* = 8). *Gapdh* mRNA levels were used for normalization. (**E**) Mature/unspliced immature transcript ratios were calculated using the results in (**B**,**D**). Levels in WT mice are set to 1. Values in graphs represent means ± sem. * *p* < 0.05, *** *p* < 0.001. ns (not significant).

**Figure 5 ijms-21-09319-f005:**
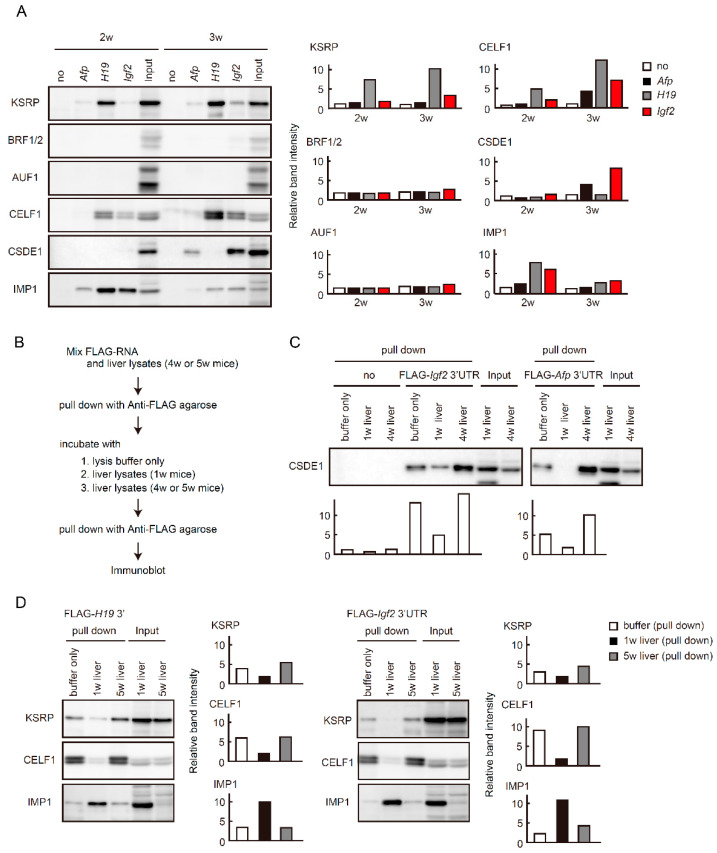
RBPs bind to 3′ regions of *Afp, H19*, and *Igf2* transcripts in a developmental stage-specific manner. (**A**) FLAG-RNAs were incubated with liver lysates from 2 w or 3 w mice. FLAG-RNA-protein complexes were purified and analyzed by immunoblot analyses. Right graphs show quantification of immunoblot data. Relative band intensities to inputs (2 w or 3 w) are calculated (*n* = 2). (**B**) Flow of competition experiments. FLAG-RNAs were incubated with liver lysates from 4 w or 5 w mice. FLAG-RNA-protein complexes were purified and then mixed with lysis buffer, liver lysates from 1 w mice, or those from 4 w or 5 w mice. After incubation, FLAG-RNA-protein complexes were purified again. (**C**,**D**) Final purified proteins as in (**B**) as well as input lysates were analyzed by immunoblot using the indicated antibodies. Graphs below blots (**C**) or on the right of blots (**D**) show quantification of immunoblot data. Relative band intensities to inputs (4 w or 5 w livers) are calculated (*n* = 2).

**Figure 6 ijms-21-09319-f006:**
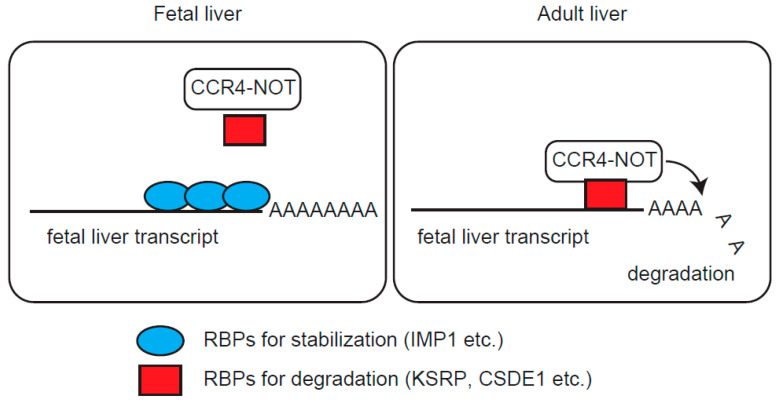
Schematic model of the postnatal decrease of fetal liver transcripts through mRNA decay. In fetal liver, RBPs for RNA stabilization (blue), such as IMP1 protect fetal liver transcripts from degradation. Expression of stabilization-related RBPs decreases during liver maturation, and instead different RBPs (red) that associate with the CCR4-NOT complex, bind to fetal liver transcripts. Consequently, fetal liver transcripts undergo CCR4-NOT complex-mediated degradation.

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
