# Peer review of "Regulation of Fetal Genes by Transitions among RNA-Binding Proteins during Liver Development"

_ijms, 2020, doi:10.3390/ijms21239319_

Round 1
Reviewer 1 Report
A nice and interesting manuscript.
Minor suggestions:
I suggest that statistical quantification be provided in figures and number of rreplicates mentioned in M&M For example I would have appreciated seeing them in:
Figure 1A, the difference between the drop in the Afp, Igf2, and H19 expression is clear between cnot3flox than Cnot3LKO. As is the decrease in Cnot3flox the decrease in Less clear, if Cnot3LKO scarcely decrease. Adding statistical data for the difference for example between 2 and 3 weeks would help. Alternatively, rewording that the decrease is significantly lee than that observed for Cnot3flox
Fig 3B
Figure 5 in general and particular 5A, in order to enforce variations for example for Afp binding.
Author Response
Reviewer 1
I suggest that statistical quantification be provided in figures and number of replicates mentioned in M&M For example I would have appreciated seeing them in:
Figure 1A, the difference between the drop in the Afp, Igf2, and H19 expression is clear between cnot3flox than Cnot3LKO. As is the decrease in Cnot3flox the decrease in Less clear, if Cnot3LKO scarcely decrease. Adding statistical data for the difference for example between 2 and 3 weeks would help. Alternatively, rewording that the decrease is significantly lee than that observed for Cnot3flox
We agree with this comment. It seems incorrect to say that fetal liver transcripts scarcely decreased in livers from Cnot3-LKO mice. As the reviewer suggested, we reworded this statement to say that the decrease of fetal liver transcripts in livers from Cnot3-LKO mice was significantly less than that observed in livers from control mice (page 3, line 100-102).
Fig 3B
The blots shown in Figure 3B are two representatives from three different samples. We added quantification data and performed statistical analyses.
Figure 5 in general and particular 5A, in order to enforce variations for example for Afp binding.
We performed two independent experiments shown in Figure 5. We quantified band intensities and calculated relative intensities. We showed means of the relative band intensities as graphs. We did not perform a statistical analysis because a sample size of 2 was insufficient to yield a meaningful result.
Reviewer 2 Report
In the current manuscript Suzuki et al. analyzed the post-transcriptional regulation of gene expression in liver mediated by different RNA-binding proteins. The role of post-transcriptional regulation in organism development is becoming more and more evident and is gaining a lot of attention recently; in this regard studied topic will be of interest to the broad audience of International Journal of Molecular Sciences. Authors demonstrated that during liver development stabilizing factors, such as IMP1, get replaced by destabilizing factors CSDE1, KSRP and CELF1, what leads to the decay of several liver-specific mRNAs. Overall, it is a very interesting paper with rigorous and comprehensive characterization of the studied mechanism.
I have only several minor comments:
- In the figure 4D authors analyzed the levels of unspliced mRNAs. The strong upregulation of igf2 is detected in IMP1 transgenic liver. However, in the text they say that levels are comparable with wild-type. Statistical analysis needs to be added to all experiments in this section.
- In the section 2.4. authors refer to the expression of Imp1 from the transgene as to “exogenous” IMP1 expression. I think saying that "IMP1 expressed from the transgene" will be more appropriate in this context.
- In the discussion authors should leave a room for other RBPs that may play a role in the regulation of gene expression in the liver, besides the ones studied in this paper. It is now well accepted that hundreds of different RBPs bind to mRNA transcripts having combinatorial effect on mRNA fate.
Author Response
Reviewer 2
I have only several minor comments:
In the figure 4D authors analyzed the levels of unspliced mRNAs. The strong upregulation of igf2 is detected in IMP1 transgenic liver. However, in the text they say that levels are comparable with wild-type. Statistical analysis needs to be added to all experiments in this section.
We had already performed statistical analyses in the quantification results. There was no significant difference (p=0.14) in levels of unspliced Igf2 transcripts between wild-type and IMP1-transgenic livers. This was due to large individual variation. The statement in the text is correct. To avoid confusion, we added ns (not significant) to bars in Figure 4 when there was no statistically significant difference.
In the section 2.4. authors refer to the expression of Imp1 from the transgene as to “exogenous” IMP1 expression. I think saying that "IMP1 expressed from the transgene" will be more appropriate in this context.
We appreciate this comment. Accordingly, we replaced the description “exogenous” IMP1 expression” with "IMP1 expressed from the transgene" in page 6, line 210-211.
In the discussion authors should leave a room for other RBPs that may play a role in the regulation of gene expression in the liver, besides the ones studied in this paper. It is now well accepted that hundreds of different RBPs bind to mRNA transcripts having combinatorial effect on mRNA fate.
We agree that other RBPs, beyond those studied in this paper, probably also play a critical role in regulation of fetal liver transcripts. We have now mentioned this possibility in the Discussion (page 10, line 311-314).
Reviewer 3 Report
This paper reported functions of several RNA binding proteins to fetal liver transcripts such as Afp, H19 and Igf2. Especially, the author investigated the relationship between polyA tail elongation and IMP1, which is a RNA binding protein and one of the IMP family proteins regulating RNA. Moreover, KSRP, CELF1 and CSDE1 bound to Afp, H19 and Igf2 and these bindings depend on the liver maturation.
Regulation mechanism of RNA stability related with liver development is very interesting and the functions of RNA binding proteins are generally important to understand each mechanism of developments and a lot of diseases. I recommend this paper will publish.
Author Response
Thank you very much for reviewing our manuscript.